**Subject Category:**
Biology (whole organism)

ecology/computational biology/environmental science

*Fraxinus excelsior*, ash, ash dieback disease, individual-based model, SORTIE, *Hymenoscyphus fraxineus*

**Author for correspondence:**
Matthew R. Evans
e-mail: mrevans@hku.hk

# Will natural resistance result in populations of ash trees remaining in British woodlands after a century of ash dieback disease?

## Matthew R. Evans

School of Biological Sciences, Kadoorie Biological Sciences Building, The University of Hong Kong, Pok Fu Lam Road, Hong Kong, People's Republic of China

MRE, 0000-0002-5630-7621

Novel pests and diseases are becoming increasingly common, and often cause additional mortality to host species in the newly contacted communities. This can alter the structure of the community up to, and including, the extinction of host species. In the last 20 years, ash dieback (ADB) disease has spread into Europe from East Asia. It has caused substantial mortality in ash tree (*Fraxinus excelsior* L.) populations. However, a proportion of the individuals in most populations appear to be less susceptible to ADB and resistance seems to have high heritability. These observations have led to suggestions that ash populations may be sustainable after the disease. In order to test this hypothesis, I modified an existing model of UK woodland (parametrized for Wytham Woods, Oxfordshire) to take into account the impact of ADB and allowed offspring to inherit resistance traits from their parent. The results suggest that ash populations would still exist in 100 years, but at lower levels than they are currently. For example, when the initial proportion of resistant individuals is about 10% and heritability of resistance is 0.5, then the population of ash falls to about one-third of present levels. The proportion of individuals initially resistant to ADB had a larger effect on population size after 100 years than the heritability of resistance. The fact that the initial size of the resistant population is important to achieve a high population size in the presence of ADB suggests that a selective breeding programme with the intention of augmenting the natural ash populations would be beneficial.

# 1. Introduction

Anthropogenically induced environmental changes (e.g. climate change, accessible long-distance transport, trade in living organisms and natural products) are increasing the frequency of introduced and native pests and pathogens [1–3]. This increase in pests and diseases is having a major impact on native species, causing additional mortality and/or reduced vitality [4]. It seems likely that long-lived, slow-reproducing species will be particularly susceptible to novel pathogens. Forest trees, for example, have been shown to be at an increasing risk from pests and pathogens that are changing their distributions as a result of both climate change and of being introduced to new areas via trade and transport [2,5–8]. The additional mortality and morbidity that results from these novel pests and pathogens can have substantial impacts on forest structure [7,8]. In 2015, *ca* 100 million hectares (about 3%) of forest globally were estimated to have been affected by pests and diseases [9].

Although for many novel pests and diseases, an impacted host population might be expected to have high susceptibility with few evolved mechanisms of resistance, not all outbreaks result in the deaths of all individuals in the population. Plants have immune mechanisms that allow them to defend themselves against pathogens; for a recent review, see [10]. Some individuals may also have traits which allow them to resist, or escape, a particular disease; for example, early leaf senescence in the autumn seems to be associated with reduced susceptibility to ash dieback (ADB) disease [11]. Similarly, resistance to Dutch elm disease seems to be associated with early bud burst in the spring [12]. A critical question in such a situation is whether there are sufficient numbers of resistant individuals, and a sufficiently high probability of offspring inheriting traits that confer resistance, to allow population persistence in the medium and long term.

ADB is caused by a fungus (*Hymenoscyphus fraxineus*, Baral *et al.*, 2014) which is native to East Asia and has colonized Europe in the last 20 years [13,14]. Within Europe, it was first reported from Poland in the early 1990s and has since spread across the continent [15]. ADB has been shown to cause significant mortality of ash trees (*Fraxinus excelsior* L.). A recent meta-analysis of the literature suggests that about 60% of ash trees in natural woodlands will die as a result of ADB, with a higher figure in plantations [16]. The existence of a sub-set of the individuals in the population which is apparently resistant, or at least less susceptible, to ADB raises the possibility that some ash trees might survive the disease. In addition, several studies have shown that resistance to ADB appears to be heritable [17–26]. This is potentially a route by which a reasonably high ash tree population could be sustained after this new threat, with dead susceptible trees being replaced by resistant individuals [21].

Forest ecosystems take many years to respond to disturbances. While it is possible to use professional opinion to intuit the likely outcome of a given change in the system, computational models allow a structured approach to projecting possible future scenarios [27]. There are a number of well-established, competing models that can be used to make projections for forests (e.g. FORMIND [28], PICUS [29] and ED [30]); in this paper, I use SORTIE [31–38]. SORTIE is an individual-based, conceptually simple model, in which trees compete only for light, and grow according to species-specific functions influenced by their light environment. This version of the model has been parametrized for a lowland woodland in the UK [39,40] and modified from the original to incorporate size-based mortality so that small (and hence young) trees have a higher mortality rate than large ones [41]. In addition to providing projections of the possible future state(s) of a system, a benefit of modelling is that it forces the modeller to be explicit about the way in which phenomena are represented in the model. This is desirable as it provides a mathematical rigour to what would otherwise be informed conjecture. All models are necessarily simplifications of the system under study, but they should aim to inform the way in which a particular problem is considered.

There has been a previous model of ADB, but the authors did not consider the possibility of inheritance of resistance traits, and assumed that ADB would be a one-off mortality event lasting a decade, after which mortality rates would return to pre-ADB levels [42]. The impact of ash mortality on forest community structure was influenced by the different demographic strategies and initial population sizes of the competing species in the forest [42]. Neither of the assumptions made by this study seems reasonable now, given evidence of heritability of resistance and mortality continuing for at least 19 years. By contrast, I assume that the disease becomes an established feature of the ecosystem, and the focus is on whether heritable resistance to ADB can result in population persistence. By incorporating the impacts of ADB on individual trees and allowing offspring to inherit parental resistance traits, the impact of ADB on forest dynamics can be projected. I believe that this is the first time that heritable resistance to a disease has been included in any forest-gap model. In this respect, this model of ADB in ash populations provides some insight into what might be required to understand the likely future invasions of novel pests into ecosystems.

# 2. Material and methods

## 2.1. Modification of the forest model to include ADB

To describe the impact of the ADB on the forest, we must understand the way in which the disease affects individual trees. For any disease, there is likely to be inter-individual variation in susceptibility to the disease; in other words, some individuals in the population are susceptible, while others are less so. Once an individual tree contracts a disease, its viability will be impaired—expressed here as an impact on growth and a reduction in crown area—and it will have an increased chance of mortality.

Heritability of resistance is important; if offspring are highly likely to inherit their parents' ability to withstand the disease, then resistant phenotypes are likely to increase in frequency relatively rapidly. In this paper, I have chosen to model susceptibility as a category, so trees can be resistant, completely susceptible or have intermediate susceptibility. This is a simplification; in reality, resistance is likely to be a continuous trait. Parent trees produce offspring that stochastically have similar susceptibility to themselves or vary in susceptibility from themselves with a certain probability [43]. It is worth noting that, as far as I am aware, there have been no studies of heritability of resistance. The studies that exist have focused on the heritability of damage scores (which could be inferred to be inversely related to resistance).

## 2.2. Estimation of disease parameters

### 2.2.1. Proportion of the ash population which is susceptible to ADB

All studies to date agree that the number of trees in the population which are resistant to ADB is low. No study seems to have found individuals which are completely immune to ADB. A study in Denmark estimated that 1% of the parent trees in that population have the potential to give rise to offspring with less than 10% crown damage [44], while another suggested that about 2.5% of their ash clones could be regarded as resistant to ADB (i.e. with less than 10% crown damage [22]). In a trial conducted over 8 years, 91.5% of the initial 27 000 seedlings died, 7.8% were severely to slightly damaged and only 0.9% appeared healthy at the end of the trial [23]. A recent review of the literature on mortality due to ADB suggested that the best estimate was that in woodland situations mortality plateaued at 60%, while if plantations were also included, the total mortality rose to 78% [16]. This estimate is likely to be sensitive to the fact that few observations have been made on forests exposed to ADB for more than 15 years (the maximum being 19 years). For modelling purposes, three scenarios were examined, in which there were high, medium and low proportions of the population resistant to ADB (table 1).

### 2.2.2. Impact of ADB on growth and crown size

When an ash tree is suffering from ADB, it has reduced viability. This is most clearly illustrated by the fact that infected ash trees grow more slowly than disease-free trees. Growth in SORTIE is modelled as radial growth in the trunk of the tree—diameter at breast height (DBH) then controls the size of the rest of the tree via allometric relationships. There are relatively few assessments of changes in radial growth due to ADB. A study of adult trees reported that the radial growth of diseased trees was 35% of that of apparently disease-free trees [45]. Infected ash saplings had radial growth 61% of that of disease-free trees [18]; an earlier study on the same trees found that radial growth was reduced by up to 26% [46]. In the model used here, radial growth was depressed by 50% for highly susceptible trees and by 20% for intermediately susceptible trees.

There would be reductions in the amount of foliage on trees suffering from dieback, as is evident in illustrations of diseased trees (see, for example, fig. 3 of [47] or fig. 2 of [21]). However, quantitative data on this impact of the disease seem to be available only rarely. One study in Poland showed that crown depth was reduced by about 15% and crown radius by 15–60% (estimated from fig. 2 of [48]). Due to this uncertainty, relatively minor impacts were imposed in the model with crown radius reduced by 20% for highly susceptible trees and by 5% for intermediately susceptible trees.

### 2.2.3. Impact of ADB on mortality

ADB clearly kills trees, with reports of stands being very severely reduced. There are reports of high annual mortality rates for seedlings and saplings, but mortality seems to be less severe for older/larger trees [19,21,23,49–51]. A Polish study reported 1-year mortality rates of 5.5% for 2–5-year-old trees, 8.0% for 6–10-year-old, 14% of 11–20-year-old trees and 0% of trees older than 20 years [52].

**Table 1.** Proportion of resistant and susceptible phenotypes in the population and the associated probabilities of a parent tree in any ADB susceptibility category producing offspring of each ADB susceptibility category.

| scenario | parent ADB category | % in population | offspring ADB category | | |
| --- | --- | --- | --- | --- | --- |
| | | | susceptible (%) | intermediately susceptible (%) | resistant (%) |
| low | susceptible | 90 | 92 | 7 | 1 |
| | intermediately susceptible | 9 | 82 | 15 | 3 |
| | resistant | 1 | 75 | 20 | 5 |
| medium | susceptible | 80 | 87 | 9 | 4 |
| | intermediately susceptible | 15 | 75 | 15 | 10 |
| | resistant | 5 | 70 | 17 | 13 |
| high ($h^2 =$ 0.5) | susceptible | 60 | 75 | 18 | 7 |
| | intermediately susceptible | 30 | 60 | 24 | 16 |
| | resistant | 10 | 53 | 27 | 20 |
| high ($h^2 =$ 0.7) | susceptible | 60 | 77 | 16 | 7 |
| | intermediately susceptible | 30 | 56 | 26 | 18 |
| | resistant | 10 | 46 | 29 | 25 |
| high ($h^2 =$ 0.3) | susceptible | 60 | 73 | 19 | 8 |
| | intermediately susceptible | 30 | 64 | 23 | 13 |
| | resistant | 10 | 60 | 25 | 15 |

In Norway, annual mortalities of 25% for trees smaller than 50 mm DBH, 8.7% for trees with DBH 50–110 mm and 2.3% for larger trees were reported [49]. Meanwhile, a French study found mortality rates of 35% $yr^{-1}$ for trees with DBH less than 50 mm, 10–11% $yr^{-1}$ for trees with DBH 50–250 mm and 3.2% $yr^{-1}$ for larger trees [50]. In a modelling exercise, rates of mortality of 3–50% a year were used for the first 10 years of an outbreak with zero additional dieback mortality thereafter; these seem to be low relative to the observations [42]. It is worth noting that these figures are total mortality and do not distinguish between ADB-induced mortality and other causes, although the latter is typically very small [41]. In this model, I increased mortality for susceptible and intermediately susceptible trees by adding an ADB annual probability of death ($P_{ADB}$), such that

$$P_{ADB} = \frac{a}{(1 + 10^{b(c-t)})},$$

where $a$ is the total chance of death due to ADB, $b$ is the slope of the curve, $c$ is the point of inflection and $t$ is years since infection. For susceptible trees, $a$ was given the value 1 (i.e. all susceptible individuals will eventually die of ADB), $b = 0.25$ and $c = 11$ years [16]. For intermediately susceptible individuals, $a = 0.9$, $b = 0.1$ and $c = 22$ as the disease was assumed to be less severe in such individuals. The time parameter ($t$) was the age of the tree if the individual was the result of reproduction during the model run (i.e. it was assumed to have been infected when it was a seedling) or the time since the start of the simulation for trees that were part of the population that started the simulation (i.e. individuals were assumed to have become infected as the simulation started).

The preceding sections have described the impact that ADB has on an ash tree; in the context of the model, these will simultaneously reduce the competitive ability of the tree (crown size reduces), reduce its growth rate (both radial and height) and increase its chances of mortality. This will mean that more light will penetrate the canopy of infected trees increasing light availability to smaller trees in the understorey, allowing them to have higher growth rates than they might otherwise have had.

### 2.2.4. Heritability of resistance

The possibility of resistance traits being heritable has received much attention. Broad sense heritability estimates (i.e. the proportion of the phenotypic variation that is due to genetic effects, plus maternal

effects, etc.) have been reported as ranging from 0.1 to 0.65 [22,24–26]. Narrow sense heritability (i.e. the proportion of the phenotypic variation that is due to additive genetic variance, $h^2$) estimates have been reported in the range 0.37–0.53 [19,23,44]. See table 1 [53] for a summary of the information to date. For this paper, I have assumed that $h^2 = 0.5$. A high heritability does suggest that offspring are likely to inherit resistance status. It is worth noting that all estimates of heritability have come from managed plantations and so may be relatively high compared to that which might be found in a woodland with less management [54].

### 2.2.5. Estimating offspring characteristics from parental characteristics

For modelling purposes, it is desirable to determine the disease resistance characteristics of offspring produced by parents in different disease resistance categories. We can approach this via the response to selection ($R$) (i.e. the deviation of offspring phenotypes from the population mean) in the offspring generation, which is given by

$$R = h^2 S,$$

where $S$ is the selection differential (i.e. the deviation of the mean phenotypic value of the parents from the population mean) [55]. In a field population of trees, all offspring will be the result of open pollination and so the probability of an offspring inheriting its mother's phenotype will be reduced from that which would be expected if there were assortative mating:

$$R = \frac{1}{2} h^2 S_f,$$

where $S_f$ is the selection differential of mother trees [55]. We can make use of the fact that

$$S_f = i_f \sigma_{pf},$$

where $i_f$ is the intensity of selection on females (estimated from the proportion of the population that is breeding) and $\sigma_{pf}$ is the population phenotypic standard deviation [55]. If 1% of ash trees are resistant to ADB, then $i_f = 2.665$ [55]. The standard deviation is 28.7–32.5, as the variance of percentage damage scores is reported as 826–1056 [22]. For modelling purposes, high values are associated with resistant individuals and low values with susceptible ones. This is the reverse of the usual presentation and employed here because what is of interest is resistance to disease, which would presumably be inversely correlated with damage score. Therefore, for resistant trees $S_f = 79.95$ (if we take the mid-point of standard deviation range to be 30), and $R = 19.99$. For the 10% most resistant trees, using the same logic, $S_f = 53.25$ and $R = 13.31$. Therefore, for the intermediate susceptibility trees (which are those between 10 and 1% most resistant), $R = 12.57$. For the most susceptible 90%, $S_f = -5.91$, and $R = -1.48$. This suggests that the mid-point of the offspring of resistant trees would be 19.99 above the population mean, while that of intermediately susceptible individuals would be 12.57 above, and susceptible trees 1.48 below the mean. The mean of the percentage damage score in [22] ranges from 24 to 56. If we take the overall mean to be 40, then this suggests mid-points of 60 for the offspring of resistant trees, 52.6 for offspring of intermediately susceptible trees and 38.5 for those of susceptible trees. If phenotypes follow a normal distribution, and the mean and variance are the same as those in [22], then the most resistant 1% of trees will be those greater than 2.326 s.d. from the mean, susceptible trees will be less than 1.341 s.d. below the mean, and intermediately susceptible trees will be between these boundaries. Therefore, if the offspring of resistant trees have a mean (±s.d.) damage score of 60 ± 30, then we should expect 5% of them to be resistant, while 20% will be intermediately susceptible, and the remaining 75% susceptible (from the probability of a value exceeding one of these thresholds if drawn from a normal distribution with the given mean and s.d.). Similarly, for an intermediately susceptible parent, we should expect 3% resistant offspring, 15% intermediately susceptible and 82% susceptible, while for a susceptible tree the figures would be 1% resistant, 7% intermediately susceptible and 92% susceptible. The full set of transition probabilities used here are shown in table 1.

## 2.3. Model simulations

I ran experimental scenarios which had high, medium and low proportions of ash trees resistant to ADB, all of which assumed $h^2 = 0.5$. In addition, the high proportion of ADB-resistant ash scenario was repeated using both $h^2 = 0.3$ and 0.7. The parameters used for each scenario are shown in table 1. Each scenario

was run 10 times and the projected number of individual trees recorded in each simulated year for 97 years. Models were initiated with tree sizes and densities calculated from the Wytham Woods ForestGEO plot [56]. Ash trees were allocated to the three resistance categories at random in the proportions required by the scenario at the start of each simulation. The simulated plot size was $500 \times 300$ m and contained 16 986 trees at the start of each simulation. I also ran a baseline scenario in which all parameters and starting conditions were identical, except for those relating to ADB. All individual ash trees in the baseline scenario had the characteristics of resistant ash in the experimental models.

SORTIE has been parametrized for the eight commonest species at our study site (Wytham Woods, Oxfordshire, UK): sycamore (*Acer pseudoplatanus* L.); European ash (*Fraxinus excelsior* L.); pedunculate oak (*Quercus robur* L.); European beech (*Fagus sylvatica* L.), common hazel (*Corylus avellana* L.); common hawthorn (*Crataegus monogyna* Jacq.), field maple (*Acer campestre* L.); and birch (*Betula* L. spp.). The numbers of each of these species are recorded at the end of each simulated year for every run. The parametrization and data are described here [39].

In this paper, which is based on the results of model simulations, the results of statistical tests have not been reported. This is because they would essentially be meaningless. The size of a statistic and its *p*-value depends on statistical power, which is determined by replication. As replication is under the control of the experimenter, then it seems inappropriate to rely on it to inform conclusions. Instead, I have relied on interpreting differences between scenarios [57].

# 3. Results

## 3.1. Impact on the ash population

The number of ash trees in the modelled forest is shown in figure 1. As would be expected, the size of the ash population drops rapidly as susceptible trees succumb to ADB. When there is a relatively large proportion of the population resistant to ADB, the ash population is sustained at about a third of what it was at the beginning of the simulation. However, this is small compared to what would be expected in the absence of ADB, as the baseline simulation suggests that a large increase in ash trees would have been expected. When the proportion of resistant individuals is low, the ash population as a whole becomes extremely small—to the point where in some instances it is one of the rarest species of the eight examined here. In the baseline scenarios, ash is by far the most abundant species in the forest.

The impact of heritability is similar to, but smaller than, the effect of the proportion of the population resistant. When heritability of resistance is 0.3, the final number of ash trees is about 15% lower than when heritability is 0.5 and 30% below in the scenario in which heritability is high (0.7). This contrasts with their being 60% fewer in the final population when 5% of the initial population is ADB resistant, compared to when there are 10% resistant individuals at the start, and 90% fewer when there are 1% resistant individuals at the start.

## 3.2. Impact on the community structure of the forest

The mean number of individuals at the end of the simulation runs of all the species considered here are shown in figure 2. The reduction in the number of ash trees is mitigated to some extent by an increase in the number of sycamores, although counterintuitively the scenario with the greatest number of sycamore is the one in which there is ADB, but the highest number of ash trees remain in the population. Both oak and hazel have higher populations when ADB is impacting the ash population than under baseline conditions. The size of the hazel population is notably more variable than those of other species and is highest when the population of ash is lowest.

# 4. Discussion

The heritability of resistance to ADB has frequently been cited as a possible long-term solution to the impact of the disease on European forests, with the prospect that naturally resistant individuals would remain and their offspring gradually recolonize the forests [19–24,44,51]. This is coupled by some authors with the suggestion of breeding-resistant lines of ash [15,17,20,23–25,58–61]. My results suggest that natural resistance could lead to population persistence, but, unless the fraction of the population of ash that is resistant to ADB is reasonably high, the overall ash population remains extremely small. This is because, even when there is high heritability, offspring are not clones of their

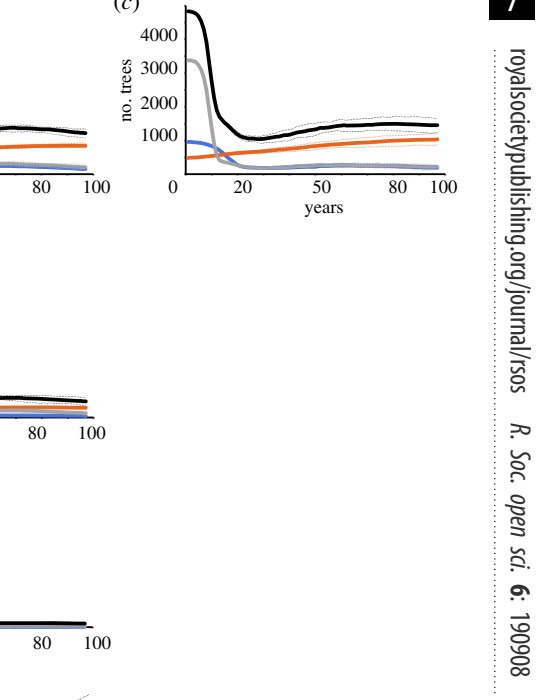

**Figure 1.** The mean number of ash trees per year. (*a*) heritability of ADB resistance 0.3, 10% ash trees resistant to ADB; (*b*) heritability of ADB resistance 0.5, 10% ash trees resistant to ADB; (*c*) heritability of ADB resistance 0.7, 10% ash trees resistant to ADB; (*d*) heritability of ADB resistance 0.5, 5% ash trees resistant to ADB; (*e*) heritability of ADB resistance 0.5, 1% ash trees resistant to ADB (orange = resistant, grey = susceptible, blue = intermediately susceptible, black = total). (*f*) The situation under baseline conditions. Solid lines represent the mean; dotted lines give 5 and 95% percentiles.

mothers. Ash are wind pollinated, and so resistant mother trees will receive pollen from a random selection of the reproductive trees in the forest, despite the fact that it is likely that susceptible, diseased individuals will produce less pollen (although the strength of this impact is not currently known). Therefore, only a relatively low number of the offspring of resistant mothers are themselves resistant. Therefore, ADB susceptible individuals are being produced even after most of the mortality due to ADB has occurred, and a susceptible population of trees persists. This will contribute to the continued persistence of the disease in the population.

One feature not considered in the model presented here is that the selection differential will change throughout the period considered. This is partly because intense selection will reduce the genetic variance in the offspring population [55] and also because as susceptible individuals die the proportion of the pollen in the air that originated from resistant fathers will rise and so the probability of resistant offspring being produced will increase. It is likely that both these effects will result in there being a higher population of resistant ash than suggested here. However, the model considers a relatively short period of time over which the number of ash generations is low. Ash grow at rates in the model that mean they only reach the size at which they would be considered capable of reproduction after at least 15 years, and usually much more (which is consistent with age at first reproduction in nature [62]). Therefore, a model run is a maximum of about six generations, but more usually two or three.

The total amount of ash pollen will also decrease as susceptible trees die. A reduction in ash pollen was detected in Austria following ADB appearing [63]. Whether a reduction in the amount of ash pollen in the air will affect the chances of viable ash producing seeds is unclear. The argument in the previous paragraph suggests that although there might be less ash pollen available to pollinate female flowers, the proportion which originates from resistant fathers is likely to increase. The overall impact of these two

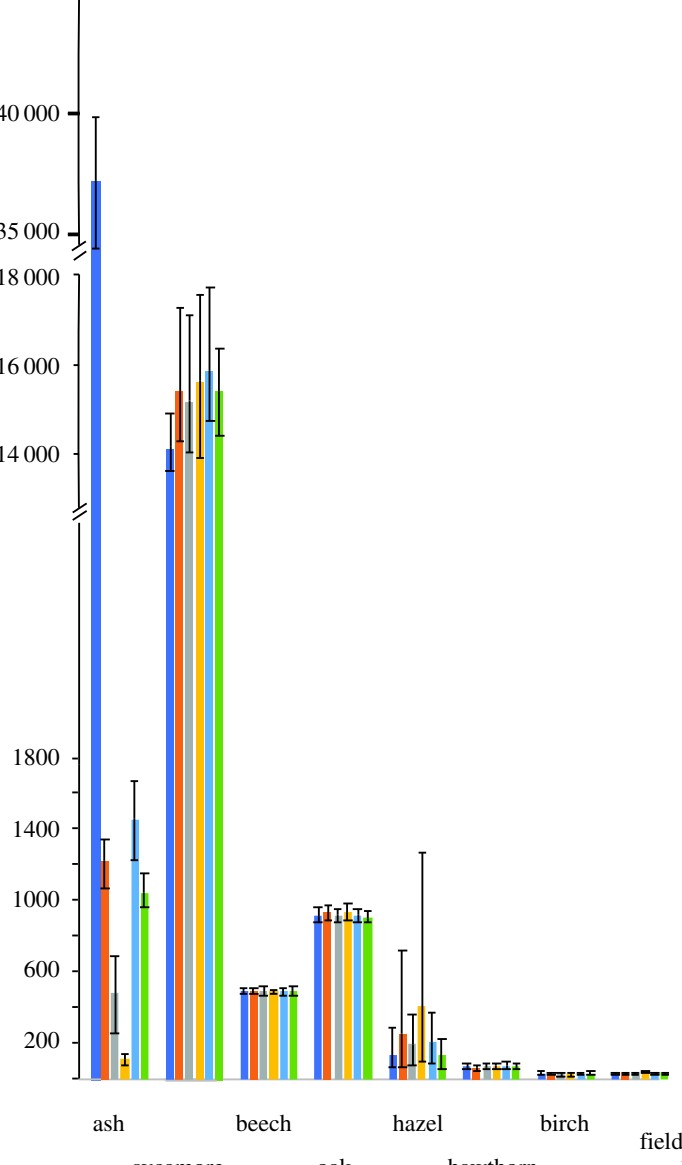

**Figure 2.** The mean number of individuals at the end of 97 simulated years of the eight tree species under the different scenarios. Bars show the mean number of trees; error bars are 5 and 95% percentiles. Dark blue bars show baseline scenario with no ADB; brown bars high proportion of ash resistant to ADB, $h^2 = 0.5$; grey medium proportion of ash resistant to ADB, $h^2 = 0.5$; yellow low proportion of ash resistant to ADB, $h^2 = 0.5$; light blue bars high proportion of ash resistant to ADB, $h^2 = 0.7$; green bars high proportion of ash resistant to ADB, $h^2 = 0.7$. Note vertical axis is broken at two points.

effects on the number of ash seedlings and the distribution of resistant and susceptible phenotypes in the offspring generation is hard to predict. A recent paper demonstrated that the majority of offspring in a patch were produced by reproduction within that patch, but that there was also a significant amount of pollen reaching the patch that originated from outside the patch [62]. Therefore, it seems unlikely that ash are pollen limited; even if many local trees are diseased, there should be sufficient pollen for fertilization from trees further afield. The distribution of ADB in the environment from which pollen is drawn is likely to be an important consideration as it would affect the relative abundances of the resistant and susceptible pollen genotypes arriving at a female and the mixture of phenotypes among the seeds.

The overall conclusion from this study is that ADB will result in a substantial fall in the number of ash in the forest, which is unsurprising. However, it is clear that the population of ash that persists depends more on the proportion of the population that is resistant to ADB than whether resistance has a high heritability. As has been suggested by other authors [15,17,20,23–25,58–61], it seems reasonable that selective breeding could play a role in the conservation of ash forests. The establishment of a source of resistant individuals that could be used to boost the numbers of naturally resistant trees in the

population would help increase the chances of population sustainability. However, it is important to recognize that the pathogen is a biological agent, and is itself subject to natural selection and will evolve. It is at least conceivable that the fungus that causes ADB could evolve in response to the population of ash becoming increasingly resistant. The current hypothesis is that, probably in part, resistance (or at least the ability to escape the disease) is conferred by earlier spring leaf flushing and earlier autumn leaf senescence [11,22]. It is possible that there could be other resistance mechanisms and a genome-wide association study suggests that there are genetic associations with ADB damage, some of which are known to be associated with pathogen responses in plants [64]. If the fungus does develop a means to overcome current resistance mechanisms, then the future of the ash would be less clear. The fungus is likely to evolve virulence mechanisms faster than the ash can evolve resistance mechanisms, partly because the generation time of the ADB fungus will be much shorter than that of the ash, which will inevitably provide it with a greater adaptive capacity [61].

While ADB is the most recent novel disease to affect the ash population, there are known to be other threats. Selection pressure exerted by ADB will reduce the genetic variance of the population and may, therefore, increase its vulnerability to other pests and diseases. It is not clear whether there are resistance mechanisms that would allow ash to cope with a second novel pest or pathogen, such as the emerald ash borer (*Agrilus planipennis* Fairmaire, 1888). If resistance to one pest or pathogen is independent of resistance to another, then the effects of a series of invasions will be additive.

A secondary result of this paper is that there are projected impacts on the community structure of the forest. The baseline conditions suggest that there should be a large increase in ash in the forest; this obviously does not occur when ADB is present. The main beneficiary appears to be sycamore, which is currently the second most dominant tree in this forest, which is consistent with a previous study [42]. All scenarios with ADB present show an increase in the number of sycamores. Sycamore is known to be susceptible to dry conditions [65]. The model used here did not include any climate change effects, which in this part of the world is likely to result in an increase in the frequency of droughts [66]. Consideration of the interaction between climate change and ADB may result in a different response from sycamore than seen here.

Trees may be peculiarly vulnerable to the effects of new pests and diseases. They are large, long-lived organisms with some, but relatively limited, capacity to combat diseases [10]. In 2015, about 100 million hectares of forest globally were impacted by pests and diseases; this represents about 3% of the world's forest cover [9]. While ADB in European ash populations is simply one example, it represents a well-studied test case of the effect of a novel disease in a wild population and the ability of that population to sustain itself, the goods and services that come from it, and the wider ecosystem of which it is part. The results presented here do suggest that a reasonably sized population of ash could be sustained into the next century. But the rapid reduction in the numbers of an abundant species will have implications for the dynamics of the forest, and the remaining population will be vulnerable to additional assaults from future pests and diseases and/or the evolution of the current disease. In all likelihood, a similar outcome would be seen if we were considering any of the other common species in the forest, many of which are already facing similar challenges [67]. It seems reasonable in the face of these challenges that steps be taken to develop programmes that will screen and select for resistance traits to new pests and diseases in native species, with a view to ensuring that the resistant population is as large as possible [68].

Data accessibility. The data used to parametrize this model have been published previously [39], it can also be found here https://matthewrevans.co.uk/databases/. Code is available from the Dryad Digital Repository: https://doi.org/10.5061/dryad.5266bm8 [69].

Competing interests. I declare I have no competing interests.

Funding. There is no funding for this study.

Acknowledgements. The author would like to thank Aris Moustakas for his work on the UK version of SORTIE, and Richard Buggs for early conversations about ADB and its impact on forests. Adeline Johns-Putra gave invaluable advice that rendered the paper more comprehensible. Three referees gave helpful suggestions on earlier versions of the MS.

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
