## [Reviewer comments · Royal Society Open Science]

Review History

RSOS-190908.R0 (Original submission)

Review form: Reviewer 1

Is the manuscript scientifically sound in its present form?

Yes

Are the interpretations and conclusions justified by the results?

Yes

Is the language acceptable?

Yes

Is it clear how to access all supporting data?

No

Do you have any ethical concerns with this paper?

No

Have you any concerns about statistical analyses in this paper?

No

Recommendation?

Accept with minor revision (please list in comments)

Comments to the Author(s)

Thank you for an interesting piece of work.

I could not easily see how the data on which the models were based could be accessed although I understand they are available.

General writing style is a little loose, with many sentences starting with dependent clauses or unnecessary phrases such as 'it is necessary to...' 'I also considered...' 'this is described by..' In almost all cases these can be removed and the sentence made more concise.

Check spelling of Fraxinus (not Fraxineus) throughout.

Page 4 Line 6. Punctuation wrong in the first sentence – should not be split up by semi-colons

Page 4 line 31 to page 5 line 16. I do not think the detail about the Dutch elm disease and mountain pine beetle outbreaks really adds much. As species examples they might be mentioned in the last sentence of the introductory paragraph and the rest cut.

Page 7. Some references are given in the text rather than as numbers?

The discussion could be shortened by about 10%.

Review form: Reviewer 2

Is the manuscript scientifically sound in its present form?

Yes

Are the interpretations and conclusions justified by the results?

Yes

Is the language acceptable?

Yes

Is it clear how to access all supporting data?

No

Do you have any ethical concerns with this paper?

No

Have you any concerns about statistical analyses in this paper?

No

Recommendation?

Major revision is needed (please make suggestions in comments)

Comments to the Author(s)

The manuscript by M. Evans entitled „Can natural resistance save ash populations from ash dieback disease?“ describes a study on the development of population sizes of ash trees under

ash dieback in the next 100 years in the UK by means of simulation models. Ash dieback is a serious tree disease that causes a dramatic decline of economically and ecologically important ash tree species in Europe. The question of future development of ash populations in Europe is of importance and I think it is relevant for the readership of RSOS.

In my opinion, the study generated only two reliable results: (1) within the next 100 years (and without breeding), the degree of susceptibility in the ash population has a larger impact on population decline than the degree of heritability of the susceptibility (or resistance). (2) The tree species that will benefit the most from ash decline in the UK is probably sycamore. Honestly, I think these two results are somewhat trivial. The first, as the author admits in the discussion, is not at all surprising when considering that 100 years correspond to only 2-3 generations of ash trees. The second result is what probably every forester in the UK would have guessed without any modelling, and only confirms what Needham et al. already reported in their paper in 2016. Other results of the manuscript, such as the magnitude and speed of the population decline, are in my opinion too dependent on unknown variables to be reliable, and the author seems to agree with this.

However, I still enjoyed reading the manuscript. To be precise: rather than the results I found the methods very interesting and informative. The strength of the manuscript is the description on how presumptions and estimates for the model were made. While I do not agree with many presumptions the author made (see specific comments directly in the manuscript in the attached pdf file), reading the manuscript sharpened my eye for the significance and the interdependencies of different aspects of ash dieback, such as mortality, growth reduction and the degree of resistance, and I believe that it would have the same effect on other researchers. I thus consider the manuscript generally worthy to be published, although the editor may have another opinion.

In the introduction, very little information is given about SORTIE, which is a forest gap model that was used as a basis for this study. I think it is necessary to provide some more information, especially on how competition power of tree species is implemented in this model. It is not very clear from the description in the manuscript, if or how loss of competition power due to disease-induced weakening was taken into account in the model. In the estimation of mortality rates, I think it is problematic that tree age was not taken into account (or only taken into account as the time a tree has been exposed to the disease). In younger trees, ash dieback is much more likely to cause mortality in a given time than in older trees. Mortality of the whole population thus depends on the age distribution of the population, and will change as the age distribution changes. In the estimation of offspring characteristics, random mating between all individuals was assumed. However, when compared to healthy (resistant) individuals, diseased trees are less likely to produce offspring. There are several recent papers by Semizer-Cuming et al. on gene flow in ash, which should be considered here. If populations of ash become very small, infection pressure will very likely decrease significantly ("dilution effect"). If these issues are too complex to be implemented in the model, they should at least be explained in the method section and/or discussion section. The limitations and weaknesses of the models should be presented more clearly in the discussion.

49 specific comments can be found directly in the manuscript in the attached pdf file (Appendix A).

Review form: Reviewer 3

Is the manuscript scientifically sound in its present form?

Yes

Are the interpretations and conclusions justified by the results?

Yes

Is the language acceptable?

Yes

Is it clear how to access all supporting data?

No

Do you have any ethical concerns with this paper?

No

Have you any concerns about statistical analyses in this paper?

No

Recommendation?

Accept with minor revision (please list in comments)

Comments to the Author(s)

This is an excellent and timely study. The adaptation of the well established SORTIE model for ash dieback is a very useful contribution to the growing literature predicting the impact of this disease. It is highly relevant to the policy question of whether or not a breeding programme is needed for ash, as the MS title suggests. It is also highly relevant for estimating the total economic cost of ash dieback. Hill et al (2019, Current Biology) recently estimated the full economic cost of ash dieback in Britain at £15bn. The author of the present MS might wish to comment on how the predictions of his model would affect this estimate.

My comments below are mainly to do with clarity of presentation.

Terms like “save ash populations” and “ash populations may survive” are ambiguous. What does salvation of a population mean? Zero reduction in size? Reduction to any viable population size, however small that may be? I suggest that this could be more precisely framed.

The MS could emphasize more that the simulation is over the course of one century, in a woodland where ash has to compete with other tree species. Thus, for example, the MS title might better describe the paper if it were “How many ash trees will survive ash dieback in a British woodland during the next century?”

The approach of the SORTIE model, its parameterization on Whytham woods, and the timeframe used need to be described more fully in the MS. I suggest the abstract should mention that the model is for Whytham woods in particular, and for a 97-year timeframe. The introduction should contain a paragraph that briefly describes what the model seeks to do, how it works and what parameters it needs. It should also briefly describe the size of Whytham woods and its tree composition, and the size of the area simulated in the model.

I think that the current second and third paragraphs of the introduction could be omitted without significant detriment to the MS, as they merely give other examples of forest pests and pathogens. More relevant would be examples (if any exist) of other forest pests and pathogens for which models of impacts have been made similar to the work of the present MS. If none exist, this could be pointed out.

I suggest that Needham et al 2016 is discussed prior to the final paragraph of the introduction,

and the final paragraph of the introduction focuses more exclusively on the work reported in the present MS.

Heritability of resistance is an important parameter for the MS. As far as I am aware, no one has yet estimated heritability of mortality due to ash dieback. All the estimates of heritability are of ash dieback damage scores such as crown dieback and lesion length, as far as I am aware. If this is the case it would be worth noting. It is also worth noting that (as far as I am aware) all estimates of heritability so far have been in plantations with fairly uniform environments. In natural woodlands environmental variation is likely to be much greater, leading to lower heritability of resistance traits.

Abstract line 20/21 "These observations have LED TO SUGGESTIONS that..."

Page 7 line 57. With a strict regard to its Latin origin, the term "decimated" means "reduced by one in ten".

Page 15 "One feature not considered in the model presented here is that the selection differential is likely to change throughout the period considered. This is partly because intense selection will reduce the genetic variance in the offspring population (51), and also because as susceptible individuals die the proportion of the pollen in the air that originated from resistant fathers will rise and so the probability of resistant offspring being produced will increase. It is likely that both these effects will result in there being a higher population of resistant ash than suggested here." This is an important caveat, especially if we are interested in timeframes of over a century, which deserves more emphasis.

Page 16 "The current hypothesis is that, probably in part, resistance is conferred by earlier spring leaf flushing and earlier autumn leaf senescence (15, 26)" this is only one of several hypotheses. If true, this phenology trend may simply allow escape of trees that flush and senesce earlier than other trees - if the whole population shifts to earlier flushing and senescence, this advantage may be lost.

Figure 1. A fuller first part of the legend would be helpful to the reader.

The MS could be clearer on how the reader can access the data and code used.

In future it would be useful if the author modelled the effects of management interventions like the felling of diseased trees, or the planting of "replacement" species for ash.

Decision letter (RSOS-190908.R0)

19-Jun-2019

Dear Miss Evans,

The editors assigned to your paper ("Can natural resistance save ash populations from ash dieback disease?") have now received comments from reviewers. We would like you to revise your paper in accordance with the referee and Associate Editor suggestions which can be found below (not including confidential reports to the Editor). Please note this decision does not guarantee eventual acceptance.

Please submit a copy of your revised paper before 12-Jul-2019. Please note that the revision deadline will expire at 00.00am on this date. If we do not hear from you within this time then it will be assumed that the paper has been withdrawn. In exceptional circumstances, extensions may be possible if agreed with the Editorial Office in advance. We do not allow multiple rounds of revision so we urge you to make every effort to fully address all of the comments at this stage. If deemed necessary by the Editors, your manuscript will be sent back to one or more of the original reviewers for assessment. If the original reviewers are not available, we may invite new reviewers.

- Data accessibility

<http://datadryad.org/submit?journalID=RSOS&manu=RSOS-190908>

- Competing interests

- Authors' contributions

All submissions, other than those with a single author, must include an Authors' Contributions section which individually lists the specific contribution of each author. The list of Authors

should meet all of the following criteria; 1) substantial contributions to conception and design, or acquisition of data, or analysis and interpretation of data; 2) drafting the article or revising it critically for important intellectual content; and 3) final approval of the version to be published.

- Acknowledgements

- Funding statement

on behalf of Kevin Padian (Subject Editor)
openscience@royalsociety.org

Comments to Author:

Reviewers' Comments to Author:

Reviewer: 1

Comments to the Author(s)

Thank you for an interesting piece of work.

I could not easily see how the data on which the models were based could be accessed although I understand they are available.

General writing style is a little loose, with many sentences starting with dependent clauses or unnecessary phrases such as 'it is necessary to...' 'I also considered...' 'this is described by..' In almost all cases these can be removed and the sentence made more concise.

Check spelling of *Fraxinus* (not *Fraxineus*) throughout.

Page 4 Line 6. Punctuation wrong in the first sentence – should not be split up by semi-colons

Page 4 line 31 to page 5 line 16. I do not think the detail about the Dutch elm disease and mountain pine beetle outbreaks really adds much. As species examples they might be mentioned in the last sentence of the introductory paragraph and the rest cut.

Page 7. Some references are given in the text rather than as numbers?

The discussion could be shortened by about 10%.

Reviewer: 2

Comments to the Author(s)

The manuscript by M. Evans entitled „Can natural resistance save ash populations from ash dieback disease?“ describes a study on the development of population sizes of ash trees under ash dieback in the next 100 years in the UK by means of simulation models. Ash dieback is a serious tree disease that causes a dramatic decline of economically and ecologically important ash tree species in Europe. The question of future development of ash populations in Europe is of importance and I think it is relevant for the readership of RSOS.

In my opinion, the study generated only two reliable results: (1) within the next 100 years (and without breeding), the degree of susceptibility in the ash population has a larger impact on population decline than the degree of heritability of the susceptibility (or resistance). (2) The tree species that will benefit the most from ash decline in the UK is probably sycamore. Honestly, I think these two results are somewhat trivial. The first, as the author admits in the discussion, is not at all surprising when considering that 100 years correspond to only 2-3 generations of ash trees. The second result is what probably every forester in the UK would have guessed without any modelling, and only confirms what Needham et al. already reported in their paper in 2016. Other results of the manuscript, such as the magnitude and speed of the population decline, are in my opinion too dependent on unknown variables to be reliable, and the author seems to agree with this.

However, I still enjoyed reading the manuscript. To be precise: rather than the results I found the methods very interesting and informative. The strength of the manuscript is the description on how presumptions and estimates for the model were made. While I do not agree with many presumptions the author made (see specific comments directly in the manuscript in the attached pdf file), reading the manuscript sharpened my eye for the significance and the interdependencies of different aspects of ash dieback, such as mortality, growth reduction and the degree of resistance, and I believe that it would have the same effect on other researchers. I thus consider the manuscript generally worthy to be published, although the editor may have another opinion.

In the introduction, very little information is given about SORTIE, which is a forest gap model that was used as a basis for this study. I think it is necessary to provide some more information, especially on how competition power of tree species is implemented in this model. It is not very clear from the description in the manuscript, if or how loss of competition power due to disease-induced weakening was taken into account in the model. In the estimation of mortality rates, I think it is problematic that tree age was not taken into account (or only taken into account as the time a tree has been exposed to the disease). In younger trees, ash dieback is much more likely to cause mortality in a given time than in older trees. Mortality of the whole population thus depends on the age distribution of the population, and will change as the age distribution changes. In the estimation of offspring characteristics, random mating between all individuals was assumed. However, when compared to healthy (resistant) individuals, diseased trees are less likely to produce offspring. There are several recent papers by Semizer-Cuming et al. on gene flow in ash, which should be considered here. If populations of ash become very small, infection pressure will very likely decrease significantly (“dilution effect”). If these issues are too complex to be implemented in the model, they should at least be explained in the method section and/or discussion section. The limitations and weaknesses of the models should be presented more clearly in the discussion.

49 specific comments can be found directly in the manuscript in the attached pdf file.

Reviewer: 3

Comments to the Author(s)

This is an excellent and timely study. The adaptation of the well established SORTIE model for ash dieback is a very useful contribution to the growing literature predicting the impact of this disease. It is highly relevant to the policy question of whether or not a breeding programme is needed for ash, as the MS title suggests. It is also highly relevant for estimating the total economic cost of ash dieback. Hill et al (2019, Current Biology) recently estimated the full economic cost of ash dieback in Britain at £15bn. The author of the present MS might wish to comment on how the predictions of his model would affect this estimate.

My comments below are mainly to do with clarity of presentation.

Terms like “save ash populations” and “ash populations may survive” are ambiguous. What does salvation of a population mean? Zero reduction in size? Reduction to any viable population size, however small that may be? I suggest that this could be more precisely framed.

The MS could emphasize more that the simulation is over the course of one century, in a woodland where ash has to compete with other tree species. Thus, for example, the MS title might better describe the paper if it were “How many ash trees will survive ash dieback in a British woodland during the next century?”

The approach of the SORTIE model, its parameterization on Whytham woods, and the timeframe used need to be described more fully in the MS. I suggest the abstract should mention that the model is for Whytham woods in particular, and for a 97-year timeframe. The introduction should contain a paragraph that briefly describes what the model seeks to do, how it works and what parameters it needs. It should also briefly describe the size of Whytham woods and its tree composition, and the size of the area simulated in the model.

I think that the current second and third paragraphs of the introduction could be omitted without significant detriment to the MS, as they merely give other examples of forest pests and pathogens. More relevant would be examples (if any exist) of other forest pests and pathogens for which models of impacts have been made similar to the work of the present MS. If none exist, this could be pointed out.

I suggest that Needham et al 2016 is discussed prior to the final paragraph of the introduction, and the final paragraph of the introduction focuses more exclusively on the work reported in the present MS.

Heritability of resistance is an important parameter for the MS. As far as I am aware, no one has yet estimated heritability of mortality due to ash dieback. All the estimates of heritability are of ash dieback damage scores such as crown dieback and lesion length, as far as I am aware. If this is the case it would be worth noting. It is also worth noting that (as far as I am aware) all estimates of heritability so far have been in plantations with fairly uniform environments. In natural woodlands environmental variation is likely to be much greater, leading to lower heritability of resistance traits.

Abstract line 20/21 “These observations have LED TO SUGGESTIONS that...”

Page 7 line 57. With a strict regard to its Latin origin, the term “decimated” means “reduced by one in ten”.

Page 15 “One feature not considered in the model presented here is that the selection differential is likely to change throughout the period considered. This is partly because intense selection will reduce the genetic variance in the offspring population (51), and also because as susceptible individuals die the proportion of the pollen in the air that originated from resistant fathers will rise and so the probability of resistant offspring being produced will increase. It is likely that both these effects will result in there being a higher population of resistant ash than suggested here.” This is an important caveat, especially if we are interested in timeframes of over a century, which deserves more emphasis.

Page 16 “The current hypothesis is that, probably in part, resistance is conferred by earlier spring leaf flushing and earlier autumn leaf senescence (15, 26)” this is only one of several hypotheses. If true, this phenology trend may simply allow escape of trees that flush and senesce earlier than other trees – if the whole population shifts to earlier flushing and senescence, this advantage may be lost.

Figure 1. A fuller first part of the legend would be helpful to the reader.

The MS could be clearer on how the reader can access the data and code used.

In future it would be useful if the author modelled the effects of management interventions like the felling of diseased trees, or the planting of “replacement” species for ash.

Author's Response to Decision Letter for (RSOS-190908.R0)

See Appendix B.

RSOS-190908.R1 (Revision)

Review form: Reviewer 1

Is the manuscript scientifically sound in its present form?

Yes

Are the interpretations and conclusions justified by the results?

Yes

Is the language acceptable?

Yes

Do you have any ethical concerns with this paper?

No

Have you any concerns about statistical analyses in this paper?

No

Recommendation?

Accept as is

Comments to the Author(s)

Thank you for your efforts in the revision.

Review form: Reviewer 2

Is the manuscript scientifically sound in its present form?

Yes

Are the interpretations and conclusions justified by the results?

Yes

Is the language acceptable?

Yes

Do you have any ethical concerns with this paper?

No

Have you any concerns about statistical analyses in this paper?

Yes

Recommendation?

Accept with minor revision (please list in comments)

Comments to the Author(s)

The manuscript has significantly improved. Most of my comments and suggestions have been taken into account by the author, although sometimes somewhat desultory, but I guess that is ok. I still think that the presentation of Figure 1 needs improvement, i.e. increased font size, and figure legends would make it much easier for the reader to understand and to extract the relevant information.

I am still confused by the calculation described in page 11. The calculated mid-points are, as I understand, the mid-points of the standard deviation range of percent damage scores of offspring of susceptible and resistant trees. If this is correct, I would expect the mid-point of offspring of resistant trees to be smaller than those of offspring of intermediate and susceptible trees. Apparently, there is something that I did not understand, but I admit that this may be my personal problem.

I could not find a statement in the discussion that a strong reduction of the host population and the number of susceptible trees is likely to cause a decrease in infection pressure.

Review form: Reviewer 3

Is the manuscript scientifically sound in its present form?

Yes

Are the interpretations and conclusions justified by the results?

Yes

Is the language acceptable?

Yes

Do you have any ethical concerns with this paper?

No

Have you any concerns about statistical analyses in this paper?

No

Recommendation?

Accept with minor revision (please list in comments)

Comments to the Author(s)

The author has dealt satisfactorily with the reviewers comments. My only further suggestion is that the word "initial" is inserted before the word "proportion" on line 31 of the abstract.

Decision letter (RSOS-190908.R1)

23-Jul-2019

Dear Professor Evans:

On behalf of the Editors, I am pleased to inform you that your Manuscript RSOS-190908.R1 entitled "Will natural resistance result in populations of ash trees remaining in British woodlands after a century of ash dieback disease?" has been accepted for publication in Royal Society Open Science subject to minor revision in accordance with the referee suggestions. Please find the referees' comments at the end of this email.

The reviewers and Subject Editor have recommended publication, but also suggest some minor revisions to your manuscript. Therefore, I invite you to respond to the comments and revise your manuscript.

- Ethics statement

- Data accessibility

It is a condition of publication that all supporting data are made available either as supplementary information or preferably in a suitable permanent repository. The data

accessibility section should state where the article's supporting data can be accessed. This section should also include details, where possible of where to access other relevant research materials such as statistical tools, protocols, software etc can be accessed. If the data has been deposited in an external repository this section should list the database, accession number and link to the DOI for all data from the article that has been made publicly available. Data sets that have been deposited in an external repository and have a DOI should also be appropriately cited in the manuscript and included in the reference list.

If you wish to submit your supporting data or code to Dryad (<http://datadryad.org/>), or modify your current submission to dryad, please use the following link:
<http://datadryad.org/submit?journalID=RSOS&manu=RSOS-190908.R1>

- **Competing interests**

- **Authors' contributions**

- **Acknowledgements**

- **Funding statement**

Because the schedule for publication is very tight, it is a condition of publication that you submit the revised version of your manuscript before 01-Aug-2019. Please note that the revision deadline will expire at 00.00am on this date. If you do not think you will be able to meet this date please let me know immediately.

To revise your manuscript, log into <https://mc.manuscriptcentral.com/rsos> and enter your Author Centre, where you will find your manuscript title listed under "Manuscripts with Decisions". Under "Actions," click on "Create a Revision." You will be unable to make your

revisions on the originally submitted version of the manuscript. Instead, revise your manuscript and upload a new version through your Author Centre.

on behalf of Prof Kevin Padian (Subject Editor)
openscience@royalsociety.org

Associate Editor Comments to Author:

Thank you for making efforts to improve the paper. Only a few minor concerns remain from the reviewers - please ensure you tackle these in your revision.

Reviewer comments to Author:

Reviewer: 3

Comments to the Author(s)

The author has dealt satisfactorily with the reviewers comments. My only further suggestion is that the word "initial" is inserted before the word "proportion" on line 31 of the abstract.

Reviewer: 1

Comments to the Author(s)

Thank you for your efforts in the revision.

Reviewer: 2

Comments to the Author(s)

The manuscript has significantly improved. Most of my comments and suggestions have been taken into account by the author, although sometimes somewhat desultory, but I guess that is ok. I still think that the presentation of Figure 1 needs improvement, i.e. increased font size, and figure legends would make it much easier for the reader to understand and to extract the relevant information.

I am still confused by the calculation described in page 11. The calculated mid-points are, as I understand, the mid-points of the standard deviation range of percent damage scores of offspring of susceptible and resistant trees. If this is correct, I would expect the mid-point of offspring of resistant trees to be smaller than those of offspring of intermediate and susceptible trees. Apparently, there is something that I did not understand, but I admit that this may be my personal problem.

I could not find a statement in the discussion that a strong reduction of the host population and the number of susceptible trees is likely to cause a decrease in infection pressure.

Author's Response to Decision Letter for (RSOS-190908.R1)

See Appendix C.

Decision letter (RSOS-190908.R2)

30-Jul-2019

Dear Professor Evans,

I am pleased to inform you that your manuscript entitled "Will natural resistance result in populations of ash trees remaining in British woodlands after a century of ash dieback disease?" is now accepted for publication in Royal Society Open Science.

You can expect to receive a proof of your article in the near future. Please contact the editorial

office (openscience_proofs@royalsociety.org and openscience@royalsociety.org) to let us know if you are likely to be away from e-mail contact. Due to rapid publication and an extremely tight schedule, if comments are not received, your paper may experience a delay in publication.

on behalf of Prof Kevin Padian (Subject Editor)
openscience@royalsociety.org

Appendix A**ROYAL SOCIETY
OPEN SCIENCE****Can natural resistance save ash populations from ash
dieback disease?**

Journal:	Royal Society Open Science
Manuscript ID	RSOS-190908
Article Type:	Research
Date Submitted by the Author:	17-May-2019
Complete List of Authors:	Evans, Matthew; University of Hong Kong, School of Biological Sciences
Subject:	ecology < BIOLOGY, computational biology < CROSS-DISCIPLINARY SCIENCES, environmental science < CROSS-DISCIPLINARY SCIENCES
Keywords:	Fraxineus excelsior, ash, ash dieback disease, individual-based model, SORTIE, forest-gap model
Subject Category:	Biology (whole organism)

**Author-supplied statements**

Relevant information will appear here if provided.

***Ethics***

*Does your article include research that required ethical approval or permits?:*

This article does not present research with ethical considerations

*Statement (if applicable):*

CUST_IF_YES_ETHICS :No data available.

***Data***

*It is a condition of publication that data, code and materials supporting your paper are made publicly*
*available. Does your paper present new data?:*

My paper has no data

*Statement (if applicable):*

CUST_IF_YES_DATA :No data available.

***Conflict of interest***

I/We declare we have no competing interests

*Statement (if applicable):*

CUST_STATE_CONFLICT :No data available.

***Authors' contributions***

I am the only author on this paper

*Statement (if applicable):*

CUST_AUTHOR_CONTRIBUTIONS_TEXT :No data available.

**Can natural resistance save ash populations from ash dieback disease?**

Matthew R. Evans,

School of Biological Sciences,

Kadoorie Biological Sciences Building,

The University of Hong Kong,

Pok Fu Lam Road,

Hong Kong SAR, China

Abstract

Novel pests and diseases are becoming increasingly frequent, and often cause additional mortality to host species in the newly contacted communities. This can alter the structure of the community and/or the continued presence of impacted host species. In the last twenty years ash dieback disease (ADB) has spread into Europe from East Asia. It has caused substantial mortality in ash tree (*Fraxinus excelsior*, L.) populations. However, a proportion of the individuals in most populations appears to be less susceptible to ADB and resistance seems to have high heritability. These observations have suggested that ash populations may survive the disease. In order to test this hypothesis, I modified an existing model of UK woodland to take into account the impact of ADB, and allowed offspring to inherit resistance traits from their parent. The results suggest that ash populations could be sustainable but at lower levels than they are currently. For example, when the proportion of resistant individuals is about 10% and heritability of resistance is 0.5 then the population of ash falls to about one third of present levels. The influence of the proportion of individuals initially resistant to ADB was larger than the heritability of resistance. The fact that the size of the resistant population is important to achieving a high population size in the presence of ADB suggests that a selective breeding programme with the intention of augmenting the natural ash populations would be beneficial.

Keywords: *Fraxinus excelsior*; ash; ash dieback disease; individual-based model; SORTIE; forest-gap model

Introduction

Anthropogenically induced environmental changes; from climate change, to accessible long-distance transport, to the trade in living organisms and natural products; are increasing the frequency of introduced and native pests and pathogens (1-3). This increase in pests and diseases is having major impacts on native species, causing additional mortality and/or reduced vitality (4). It seems likely that long-lived, slow-reproducing species will be particularly susceptible to novel pathogens. Forest trees, for example, have been shown to be at increasing risk from pests and pathogens that are changing their distributions as a result both of climate change, and being introduced to new areas via trade and transport (2, 5-8). The resultant additional mortality and morbidity can have substantial impacts on forest structure (7, 8).

There are many examples of the individual and population-level impacts of outbreaks of pests and pathogens in forests (6). A well-known one would be Dutch elm disease which is caused by ascomycete fungi (*Ophiostoma* spp., Syd. & P. Syd, 1919) that arrived in Europe and North America in the mid-20th Century, probably via imported timber (9). The fungus is introduced into trees by bark beetles of the genus *Scolytus* (Geoffroy, 1762) and resulted in high mortality of several, once abundant, species of elm (*Ulmus* spp., L.) (10). Populations of affected elm species remain very low. The mountain pine beetle (*Dendroctonus ponderosae*, Hopkins, 1902) is currently having a major impact. It is native to the western pine forests of North America an ecosystem that is maintained by periodic fires, which produce extensive stands of similarly aged trees within the forest (5). Changes in climate, increased temperatures in both summer and winter along with reduced precipitation, has allowed the mountain pine beetle to expand its range further north and to

higher elevations into new forest ecosystems (11). This expansion has resulted in changes
in forest structure and function; for example, it has resulted in the conversion of the forest
from a small net carbon sink into a large net carbon source (5, 12). Outbreaks of mountain
pine beetle in North America are contributing significantly to the almost 100 million
hectares of forest globally that were estimated to have been affected by pests and diseases
in 2015 (13).

[revised manuscript text omitted]

I also considered that there would be reductions in the amount of foliage on trees
suffering from dieback. This is described by the majority of studies on the effects of the
disease, and is evident in illustrations of diseased trees (see for example Figure 3 of (45) or
figure 2 of (25)). However, quantitative data on this impact of the disease seem to be
**unavailable**. Therefore, relatively minor impacts were imposed in the model with crown
radius reduced by 20% for highly susceptible trees and by 5% for intermediately susceptible
trees.

Impact of ash dieback on mortality

Ash dieback clearly kills trees, with reports of stands being decimated. There are reports of
high seedling and sapling annual mortality rates, but mortality seems to be less severe for

older/larger trees (23, 25, 27, 46-48). Kowalski *et al.* (2005) reported one year mortality
rates of 5.5% for 2-5 year old trees, 8.0% for 6-10 year old, 14% of 11-20 year old and 0% of
trees older than 20 years (49). Timmerman *et al.* (2017) found annual mortality of 25% for
trees smaller than 50mm DBH, 8.7% for trees with DBH 50 – 110mm and 2.3% for larger

[revised manuscript text omitted]

can lead to population persistence, but unless the population of ash which is resistant to
ADB is reasonably high then the overall ash population becomes extremely small. This is
because even when there is high heritability it does not mean that offspring will be clones of
their mothers. In addition, ash are wind pollinated means that resistant mother trees will
receive pollen from a random selection of the reproductive trees in the forest. Therefore,
only a relatively low number of the offspring of resistant mothers are themselves resistant.
Therefore, ADB susceptible individuals are being produced even after most of the mortality
due to ADB has occurred, and a susceptible population of trees persists. This will contribute
to the continued persistence of the disease in the population.

One feature not considered in the model presented here is that the selection
differential is likely to change throughout the period considered. This is partly because
intense selection will reduce the genetic variance in the offspring population (51), and also
because as susceptible individuals die the proportion of the pollen in the air that originated
from resistant fathers will rise and so the probability of resistant offspring being produced
will increase. It is likely that both these effects will result in there being a higher population
of resistant ash than suggested here. However, the model considers a relatively short period
of time over which the number of ash generations is low. Ash grow at rates that mean they
only reach the size at which they would be considered capable of reproduction after **at least**
**15 years, and usually much more.** The model run therefore is a maximum of about six
generations, but more usually two or three. The total amount of ash pollen will also
decrease as susceptible trees die. A reduction in ash pollen was detected in Austria
following ADB appearing, falling ash seed harvests were also reported in the same study
(58). Whether a reduction in the amount of ash pollen in the air will affect the chances of a
viable ash producing seeds is unclear.

The overall conclusion from this study is that ADB will result in a substantial fall in
the number of ash in the forest. Whether the population of ash persists seems to depend
more on the proportion of the population which is resistant to ADB rather than resistance
having high heritability. As has been suggested by other authors (19, 21, 24, 27-29, 54-57) it
seems reasonable that selective breeding could play a role. Establishing a source of resistant
individuals that could be used to boost the numbers of naturally resistant trees in the
population would help increase the chances of population sustainability. However, it is
important to recognise that the pathogen is a biological agent and is itself subject to
**selection.** It is at least conceivable that the fungus that causes ADB could evolve in response

to the population of ash becoming increasingly resistant. The current hypothesis is that,
probably in part, resistance is conferred by earlier spring leaf flushing and earlier autumn
leaf senescence (15, 26). If the fungus does develop means to overcome current resistance
mechanisms then the future of the ash would be less clear. The fungus is likely to evolve
virulence mechanisms faster than the ash can evolve resistance mechanisms partly because
the generation time of the ADB fungus will be much shorter than that of the ash, which will
inevitably provide it with greater adaptive capacity (57).

While ADB is the most recent disease to affect the ash population, there are known
to be other threats. Selection pressure exerted by ADB will reduce the genetic variance of
the population and may therefore increase its vulnerability to other pests and diseases, for
example the emerald ash borer (*Agrilus planipennis*, Fairmaire, 1888) which has already
reached Europe. It is not clear whether there are resistance mechanisms that would allow
ash to cope with this second novel pathogen. Even if there are, and with the reasonable
assumption that resistance to ADB is independent of resistance to emerald ash borer, then
the long-term future of ash in European forests looks very uncertain.

A secondary result of this paper is that there are projected impacts on the
community structure of the forest. The baseline conditions suggest that there should be a
large increase in ash in the forest, this obviously does not occur when ADB is present. The
main beneficiary appears to be sycamore which is currently the second most dominant tree
in this forest. All scenarios with ADB present show an increase in the number of sycamore.
Sycamore is known to be susceptible to dry conditions (59). The model used here did not
include any climate change effects, which in this part of the world is likely to result in an
increase in the frequency of droughts (60). Consideration of the interaction between climate
change and ADB may result in a different response from sycamore than seen here. Of the

other species considered there were rather few patterns, although hazel seem to
sometimes show increased populations in response to a reduction in the ash population.

Trees may be peculiarly vulnerable to the effects of new pests and diseases. They are
large, long-lived organisms with some, but relatively limited, capacity to combat diseases
(14). In 2015 about 100m Ha of forest globally were impacted by pests and diseases, this
represents about 3% of the world's forest cover (13). While ADB in European ash
populations is simply one example, it represents a well-studied test case of the effect of a
novel disease in a wild population and the ability of that population to sustain itself, the
goods and service that come from it, and the wider ecosystem of which it is part. The results
presented here do suggest that it the parameters we have used are correct and if about 10%
of the ash population is resistant to ADB then a reasonable sized population of ash can be
sustained into the next century. But the rapid reduction in the numbers of an abundant
species will have implications for the dynamics of the forest, and the remaining population
will be vulnerable to additional assaults from future pests and diseases and/or the evolution
of the current disease. In all likelihood a similar outcome would be seen if we were
considering any of the other common species in the forest, many of which are already facing
similar challenges (61). It seems reasonable in the face of these challenges that steps be
taken to develop programmes that will screen and select for resistance traits to new pests
and diseases in native species with a view to ensuring that the resistant population is as
large as possible (62).

**Acknowledgements**

I would like to thank Aris Moustakas for his work on the UK version of SORTIE, and Richard

Table 1. Proportion of resistant and susceptible phenotypes in the population and the associated probabilities of a parent tree in any ash dieback susceptibility category producing offspring of each ash dieback susceptibility category.

Scenario	Parent ash dieback category	% in pop'n	Offspring ash dieback category		
			Susceptible	Intermediately susceptible	Resistant
Low	Susceptible	90%	92%	7%	1%
	Intermediately susceptible	9%	82%	15%	3%
	Resistant	1%	75%	20%	5%
Medium	Susceptible	80%	87%	9%	4%
	Intermediately susceptible	15%	75%	15%	10%
	Resistant	5%	70%	17%	13%
High $h^2 = 0.5$	Susceptible	60%	75%	18%	7%
	Intermediately susceptible	30%	60%	24%	16%
	Resistant	10%	53%	27%	20%
High $h^2 = 0.7$	Susceptible	60%	77%	16%	7%
	Intermediately susceptible	30%	56%	26%	18%
	Resistant	10%	46%	29%	25%
High $h^2 = 0.3$	Susceptible	60%	73%	19%	8%
	Intermediately susceptible	30%	64%	23%	13%
	Resistant	10%	60%	25%	15%

**Figure 1.** Mean number of ash trees per year. a) – c) the proportion of ash trees resistant to
ADB is high (10% of population), a) heritability of ADB resistance 0.3; b), d) and e)
heritability of ADB resistance 0.5; c) heritability of ADB resistance 0.7. d) the proportion of
ash trees resistant to ADB is medium (5% of population); and e) the proportion of ash trees
resistant to ADB is low (1% of population) (orange = resistant, grey = susceptible, blue =
intermediately susceptible, black = total). f) shows the situation under baseline conditions.
Solid lines represent the mean, dotted lines give 5 and 95% percentiles.

**Figure 2.** Mean number of individuals at the end of 97 simulated years of the eight tree
species under the different scenarios. Bars show mean number of trees, error bars are 5%
and 95% percentiles. Dark blue bars show baseline scenario with no ADB; brown bars high
proportion of ash resistant to ADB, $h^2 = 0.5$; grey medium proportion of ash resistant to
ADB, $h^2 = 0.5$; yellow low proportion of ash resistant to ADB, $h^2 = 0.5$; light blue bars high
proportion of ash resistant to ADB, $h^2 = 0.7$; green bars high proportion of ash resistant to
ADB, $h^2 = 0.7$. Note vertical axis is broken at two points.

References:

1. Fisher MC, Henk DA, Briggs CJ, Brownstein JS, Madoff LC, McCraw SL, et al. Emerging fungal threats to animal, plant and ecosystem health. *Nature*. 2012;484:186.
2. Brasier CM. The biosecurity threat to the UK and global environment from international trade in plants. *Plant Pathology*. 2008;57:792-808.
3. Quine C, Cahalan C, Hester A, Humphrey J, Kirby J, Moffat A, et al. Woodlands. The UK National Ecosystem Assessment Technical Report UK National Ecosystem Assessment. Cambridge, UK: UNEP-WCMC; 2011.
4. Dobson A. Climate variability, global change, immunity, and the dynamics of infectious diseases. *Ecology*. 2009;90:920-7.
5. Logan J, Powell J. Ghost forests, global warming, and the mountain pine beetle (Coleoptera: Scolytidae). *American Entomologist*. 2001;47:160-73.
6. Sturrock RN, Frankel SJ, Brown AV, Hennon PE, Kliejunas JT, Lewis KJ, et al. Climate change and forest diseases. 2011;60(1):133-49.
7. Dukes JS, Pontius J, Orwig D, Garnas JR, Rodgers VL, Brazeel N, et al. Responses of insect pests, pathogens, and invasive plant species to climate change in the forests of northeastern North America: What can we predict? This article is one of a selection of papers from NE Forests 2100: A Synthesis of Climate Change Impacts on Forests of the Northeastern US and Eastern Canada. *Canadian Journal of Forest Research*. 2009;39(2):231-48.
8. Capretti P, Thomsen IM, Kasanen R, Hietala AM, Von Weissenberg K. Forest pathogens with higher damage potential due to climate change in Europe AU - La Porta, N. *Canadian Journal of Plant Pathology*. 2008;30(2):177-95.
9. Harris E. The European White Elm, *Ulmus laevis* Pall. *Quarterly Review of Forestry*. 2017;111:263.
10. Brasier CM. Rapid Evolution of Introduced Plant Pathogens via Interspecific Hybridization: Hybridization is leading to rapid evolution of Dutch elm disease and other fungal plant pathogens. *BioScience*. 2001;51:123-33.
11. Williams DW, Liebold AM. Climate change and the outbreak ranges of two North American bark beetles. *Agr For Ent*. 2002;4:87-99.
12. Kurz WA, Dymond CC, Stinson G, Rampley GJ, Neilson ET, Carroll AL, et al. Mountain pine beetle and forest carbon feedback to climate change. *Nature*. 2008;452:987.
13. van Lierop P, Lindquist E, Sathyapala S, Franceshini G. Global forest area disturbance from fire, insect pests, diseases and severe weather events. *Forest Ecology and Management*. 2015;352:78-88.
14. Jones JDG, Dangl JL. The plant immune system. *Nature*. 2006;444:323.
15. Nielsen LR, McKinney LV, Kjaer ED. Host Phenological Stage Potentially Affects Dieback Severity after *Hymenoscyphus fraxineus* Infection in *Fraxinus excelsior* Seedlings. *Baltic Forestry*. 2017;23(1):229-32.
16. Ghelardini L. Bud Burst Phenology, Dormancy Release & Susceptibility to Dutch Elm Disease in Elms (*Ulmus* spp.). Uppsala, Sweden: Swedish University of Agricultural Sciences; 2007.
17. Gross A, Holdenrieder O, Pautasso M, Queloz V, Sieber TN. *Hymenoscyphus pseudoalbidus*, the causal agent of European ash dieback. *Mol Plant Pathol*. 2014;15:5-21.
18. Baral HO, Queloz V, Hosoya T. *Hymenoscyphus fraxineus*, the correct scientific name for the fungus causing ash dieback in

Europe. *IMA Fungus*. 2014;5:79-80.

19. Pautasso M, Aas G, Queloz V, Holdenrieder O. European ash (*Fraxinus excelsior*)
dieback - A conservation biology challenge. *Biological Conservation*. 2013;158:37-49.

20. Coker TLR, Rozypalek J, Edwards A, Harwood TP, Butfoy L, Buggs RJA. Estimating
mortality rates of European ash (*Fraxinus excelsior*) under the ash dieback (*Hymenoscyphus*
*fraxinus*) epidemic. *Plants, People, Planet*. 2019;1:48-58.

21. Enderle R, Buskamp J, Metzler B. Growth Performance of Dense Natural
Regeneration of *Fraxinus excelsior* under Attack of the Ash Dieback Agent *Hymenoscyphus*
*fraxineus*. *Baltic Forestry*. 2017;23(1):218-28.

22. Enderle R, Peters F, Nakou A, Metzler B. Temporal development of ash dieback
symptoms and spatial distribution of collar rots in a provenance trial of *Fraxinus excelsior*.
*European Journal of Forest Research*. 2013;132(5-6):865-76.

23. Lobo A, Hansen JK, McKinney LV, Nielsen LR, Kjaer ED. Genetic variation in dieback
resistance: growth and survival of *Fraxinus excelsior* under the influence of *Hymenoscyphus*
*pseudoalbidus*. *Scandinavian Journal of Forest Research*. 2014;29(6):519-26.

24. Lobo A, McKinney LV, Hansen JK, Kjaer ED, Nielsen LR. Genetic variation in dieback
resistance in *Fraxinus excelsior* confirmed by progeny inoculation assay. *Forest Pathology*.
2015;45(5):379-87.

25. McKinney LV, Nielsen LR, Collinge DB, Tomsen IM, Hansen JK, Kjaer ED. The ash
dieback crisis: genetic variation in resistance can prove a long-term solution. *Plant*
*Pathology*. 2014;63:485-99.

26. McKinney LV, Nielsen LR, Hansen JK, Kjaer ED. Presence of natural genetic resistance
in *Fraxinus excelsior* (Oleraceae) to *Chalara fraxinea* (Ascomycota): an emerging infectious
disease. *Heredity*. 2011;106(5):788-97.

27. Pliura A, Lygis V, Suchockas V, Bartkevicius E. Performance of Twenty Four European
*Fraxinus excelsior* Populations in Three Lithuanian Progeny Trials with a Special Emphasis on
Resistance to *Chalara Fraxinea*. *Baltic Forestry*. 2011;17(1):17-34.

28. Pliura A, Marciulyniene D, Bakys R, Suchockas V. Dynamics of Genetic Resistance to
*Hymenoscyphus pseudoalbidus* in Juvenile *Fraxinus excelsior* Clones. *Baltic Forestry*.
2014;20(1):10-27.

29. Stener LG. Genetic evaluation of damage caused by ash dieback with emphasis on
selection stability over time. *Forest Ecology and Management*. 2018;409:584-92.

30. Stener LG. Clonal differences in susceptibility to the dieback of *Fraxinus excelsior* in
southern Sweden. *Scandinavian Journal of Forest Research*. 2012;28.

31. Pacala SW, Canham CD, Saponara J, Silander JA, Kobe RK, Ribbens E. Forest models
defined by field measurements: estimation, error analysis and dynamics. *Ecological*
*Monographs*. 1996;66:1-43.

32. Purves D, Pacala S. Predictive models of forest dynamics. *Science*.
2008;320(5882):1452-3.

33. Strigul N, Pristinski D, Purves D, Dushoff J, Pacala S. Scaling from trees to forests:
tractable macroscopic equations for forest dynamics. *Ecological Monographs*.
2008;78(4):523-45.

34. Purves DW, Lichstein JW, Strigul N, Pacala SW. Predicting and understanding forest
dynamics using a simple tractable model. *Proceedings Of The National Academy Of Sciences*
*Of The United States Of America*. 2008;105:17018-22.

- 35. Coomes DA, Kunstler G, Canham CD, Wright E. A greater range of shade-tolerance
- niches in nutrient-rich forests: an explanation for positive richness–productivity
- relationships? *Journal of Ecology*. 2009;97(4):705-17.
- 36. Forsyth DM, Wilson DJ, Easdale TA, Kunstler G, Canham CD, Ruscoe WA, et al.
- Century-scale effects of invasive deer and rodents on the dynamics of forests growing on
- soil of contrasting fertility. *Ecological Monographs*. 2015;85:157-80.
- 37. Kunstler G, Allen RB, Coomes DA, Canham CD, Wright EF. SORTIE/NZ Model
- Development: Landcare Research New Zealand Ltd; 2011.
- 38. Kunstler G, Coomes DA, Canham CD. Size-dependence of growth and mortality
- influence the shade tolerance of trees in a lowland temperate rain forest. *Journal of*
- *Ecology*. 2009;97:685-95.
- 39. Evans MR, Moustakas A, Carey G, Malhi Y, Butt N, Benham S, et al. Allometry and
- growth of eight tree taxa in United Kingdom woodlands. *Scientific Data*. 2015;2:150006.
- 40. Evans MR, Moustakas A. A comparison between data requirements and availability
- for calibrating predictive ecological models for lowland UK woodlands: learning new tricks
- from old trees. *Ecology and Evolution*. 2016.
- 41. Needham J, Merow C, Butt N, Malhi Y, Marthews TR, Morecroft M, et al. Forest
- community response to invasive pathogens: the case of ash dieback in a British woodland.
- *Journal of Ecology*. 2016;104(2):315-30.
- 42. Moustakas A, Evans MR. Integrating evolution into ecological modelling:
- accommodating phenotypic changes in agent based models. *PloS One*. 2013;8:e71125.
- 43. Kjaer ED, McKinney LV, Nielsen LR, Hansen LN, Hansen JK. Adaptive potential of ash
- (*Fraxinus excelsior*) populations against the novel emerging pathogen *Hymenoscyphus*. *Evol*
- *Appl*. 2011;5:219-28.
- 44. Vacek S, Vacek Z, Bulusek D, Putalova T, Sarginci M, Schwarz O, et al. European Ash
- (*Fraxinus excelsior* L.) Dieback: Disintegrating forest in the mountain protected areas, Czech
- Republic. *Austrian Journal of Forest Science*. 2015;132(4):203-23.
- 45. Kirisits T, Freinschlag C. Ash dieback caused by *Hymenoscyphus pseudoalbidus* in a
- seed plantation of *Fraxinus excelsior* in Austria. *J Agric Ext Rural Dev*. 2012;4:184-91.
- 46. Timmermann V, Nagy NE, Hietala AM, Borja I, Solheim H. Progression of Ash Dieback
- in Norway Related to Tree Age, Disease History and Regional Aspects. *Baltic Forestry*.
- 2017;23(1):150-8.
- 47. Marçais B, Husson C, Cael O, Dowkiw A, Saintonge FX, Delahaye L, et al. Estimation of
- Ash Mortality Induced by *Hymenoscyphus fraxineus* in France and Belgium. *Baltic Forestry*.
- 2017;23(1):159-67.
- 48. Pliura A, Lygis V, Marciulyniene D, Suchockas V, Bakys R. Genetic variation of
- *Fraxinus excelsior* half-sib families in response to ash dieback disease following simulated
- spring frost and summer drought treatments. *Iforest-Biogeosciences and Forestry*.
- 2016;9:12-22.
- 49. Kowalski T, Lukomska A. Badania nad zamieraniem jesionu (*Fraxinus excelsior* L.) w
- drzewostanach Nadleśnictwa Włoszczowa. *Acta Agrobotanica*. 2005;58:429-40.
- 50. Moustakas A, Evans MR. Effects of growth rate, size, and light availability on tree
- survival across life stages: a demographic analysis accounting for missing values and small
- sample sizes. *BMC Ecology*. 2015;15.
- 51. Falconer DS, MacKay TFC. *Introduction to Quantitative Genetics*. 4th ed. Harlow,
- U.K.: Pearson Education Ltd; 1996. 464 p.

- 52. Anderson-Teixeira KJ, Davies SJ, Bennett AC, Gonzalez-Akre EB, Muller-Landau HC, Joseph Wright S, et al. CTFS-ForestGEO: a worldwide network monitoring forests in an era of global change. *Global Change Biology*. 2015;21(2):528-49.
- 53. Wilson White J, Rassweller A, Samhoury JF, Stier AC, White C. Ecologists should not use statistical significance tests to interpret simulation model results. *Oikos*. 2013;123:385-8.
- 54. Fussi B, Konnert M. Genetic analysis of European common ash (*Fraxinus excelsior* L.) populations affected by ash dieback. *Silvae Genetica*. 2014;63(5):198-212.
- 55. Stocks JJ, Buggs RJA, Lee SJ. A first assessment of *Fraxinus excelsior* (common ash) susceptibility to *Hymenoscyphus fraxineus* (ash dieback) throughout the British Isles. *Scientific Reports*. 2017;7.
- 56. Sniezko RA, Koch J. Breeding trees resistant to insects and diseases: putting theory into application. *Biological Invasions*. 2017;19(11):3377-400.
- 57. Landolt J, Gross A, Holdenrieder O, Pautasso M. Ash dieback due to *Hymenoscyphus fraxineus*: what can be learnt from evolutionary ecology? *Plant Pathology*. 2016;65(7):1056-70.
- 58. Heinze B, Tiefenbacher H, Litchauer R, Kirisits T. Ash dieback in Austria – history, current situation and outlook. In: Vasaitis R, Enderle R, editors. *Dieback of European Ash (Fraxinus spp): Consequences and Guidelines for Sustainable Management*. Uppsala, Sweden: Swedish University of Agricultural Sciences; 2017. p. 33-52.
- 59. Morecroft MD, Stokes VJ, Taylor ME, Morison JIL. Effects of climate and management history on the distribution and growth of sycamore (*Acer pseudoplatanus* L.) in a southern British woodland in comparison to native competitors. *Forestry*. 2008;81:59-74.
- 60. Murphy JM, Sexton DMH, Jenkins GJ, Boorman PM, Booth BBB, Brown CC, et al. UKCP09 Science Report: Climate change projections. Exeter, UK.: Met Office Hadley Centre; 2009.
- 61. Cavers S. Evolution, ecology and tree health: finding ways to prepare Britain's forests for future threats. *Forestry*. 2015;88(1):1-2.
- 62. Ennos RA. Resilience of forests to pathogens: an evolutionary ecology perspective. *Forestry*. 2015;88(1):41-52.

Mean number of ash trees per year. a) – c) the proportion of ash trees resistant to ADB is high (10% of population), a) heritability of ADB resistance 0.3; b), d) and e) heritability of ADB resistance 0.5; c) heritability of ADB resistance 0.7. d) the proportion of ash trees resistant to ADB is medium (5% of population); and e) the proportion of ash trees resistant to ADB is low (1% of population) (orange = resistant, grey = susceptible, blue = intermediately susceptible, black = total). f) shows the situation under baseline conditions. Solid lines represent the mean, dotted lines give 5 and 95% percentiles.

179x155mm (300 x 300 DPI)

Mean number of individuals at the end of 97 simulated years of the eight tree species under the different scenarios. Bars show mean number of trees, error bars are 5% and 95% percentiles. Dark blue bars show baseline scenario with no ADB; brown bars high proportion of ash resistant to ADB, $h^2 = 0.5$; grey medium proportion of ash resistant to ADB, $h^2 = 0.5$; yellow low proportion of ash resistant to ADB, $h^2 = 0.5$; light blue bars high proportion of ash resistant to ADB, $h^2 = 0.7$. Note vertical axis is broken at two points.

131x228mm (300 x 300 DPI)

Appendix B

Comments to Author:

Reviewers' Comments to Author:

Reviewer: 1

Comments to the Author(s)

Thank you for an interesting piece of work.

I could not easily see how the data on which the models were based could be accessed although I understand they are available.

An explicit statement referencing a previous paper has been added on page 11.

General writing style is a little loose, with many sentences starting with dependent clauses or unnecessary phrases such as 'it is necessary to...' 'I also considered...' 'this is described by..' In almost all cases these can be removed and the sentence made more concise.

These phrases have been removed as far as possible

Check spelling of Fraxinus (not Fraxineus) throughout.

Apologies, I have done so. Although please note that the name of the fungus that produces ash dieback disease is Hymenoscyphus fraxineus.

Page 4 Line 6. Punctuation wrong in the first sentence – should not be split up by semi-colons

I have modified the sentence.

Page 4 line 31 to page 5 line 16. I do not think the detail about the Dutch elm disease and mountain pine beetle outbreaks really adds much. As species examples they might be mentioned in the last sentence of the introductory paragraph and the rest cut.

I have removed these sections as suggested.

Page 7. Some references are given in the text rather than as numbers?

These have been removed.

The discussion could be shortened by about 10%.

I have attempted to make the discussion more concise.

Reviewer: 2

Comments to the Author(s)

The manuscript by M. Evans entitled „Can natural resistance save ash populations from ash dieback disease?“ describes a study on the development of population sizes of ash trees under ash dieback in the next 100 years in the UK by means of simulation models. Ash dieback is a serious tree disease that causes a dramatic decline of economically and ecologically important ash tree species in Europe. The question of future development of ash populations in Europe is of importance and I think it is relevant for the readership of RSOS.

In my opinion, the study generated only two reliable results: (1) within the next 100 years (and without breeding), the degree of susceptibility in the ash population has a larger impact on population decline than the degree of heritability of the susceptibility (or resistance). (2) The tree species that will benefit the most from ash decline in the UK is probably sycamore. Honestly, I think these two results are somewhat trivial. The first, as the author admits in the discussion, is not at all surprising when considering that 100 years correspond to only 2-3 generations of ash trees. The second result is what

probably every forester in the UK would have guessed without any modelling, and only confirms what Needham et al. already reported in their paper in 2016. Other results of the manuscript, such as the magnitude and speed of the population decline, are in my opinion too dependent on unknown variables to be reliable, and the author seems to agree with this.

However, I still enjoyed reading the manuscript. To be precise: rather than the results I found the methods very interesting and informative. The strength of the manuscript is the description on how presumptions and estimates for the model were made. While I do not agree with many presumptions the author made (see specific comments directly in the manuscript in the attached pdf file), reading the manuscript sharpened my eye for the significance and the interdependencies of different aspects of ash dieback, such as mortality, growth reduction and the degree of resistance, and I believe that it would have the same effect on other researchers. I thus consider the manuscript generally worthy to be published, although the editor may have another opinion.

In the introduction, very little information is given about SORTIE, which is a forest gap model that was used as a basis for this study. I think it is necessary to provide some more information, especially on how competition power of tree species is implemented in this model.

A new section has been added to the last paragraph of the introduction to provide this information

It is not very clear from the description in the manuscript, if or how loss of competition power due to disease-induced weakening was taken into account in the model.

I have added a statement summarising the overall effect in the methods

In the estimation of mortality rates, I think it is problematic that tree age was not taken into account (or only taken into account as the time a tree has been exposed to the disease). In younger trees, ash dieback is much more likely to cause mortality in a given time than in older trees. Mortality of the whole population thus depends on the age distribution of the population, and will change as the age distribution changes.

The referees concerns are noted but no data exist on the impact of age on mortality, in this model size is used a proxy for age. Size-based mortality is part of the underlying model, although it is not part of the ADB-induced mortality. We have summarised the evidence for age/size effects on mortality in the model parameterisation section. But I opted to follow the analysis of Coker et al 2019.

In the estimation of offspring characteristics, random mating between all individuals was assumed. However, when compared to healthy (resistant) individuals, diseased trees are less likely to produce offspring. There are several recent papers by Semizer-Cuming et al. on gene flow in ash, which should be considered here. If populations of ash become very small, infection pressure will very likely decrease significantly (“dilution effect”). If these issues are too complex to be implemented in the model, they should at least be explained in the method section and/or discussion section. The limitations and weaknesses of the models should be presented more clearly in the discussion.

I feel that these effects are already mentioned in the second paragraph of the discussion. I have expanded this a little and added a reference to Semizer-Cuming.

49 specific comments can be found directly in the manuscript in the attached pdf file.

I thank the referee for the care taken in reviewing the article, I have modified the manuscript as suggested in all cases except for the first comment which I felt could be left in its more general form.

Reviewer: 3

Comments to the Author(s)

This is an excellent and timely study. The adaptation of the well established SORTIE model for ash dieback is a very useful contribution to the growing literature predicting the impact of this disease. It is highly relevant to the policy question of whether or not a breeding programme is needed for ash, as the MS title suggests. It is also highly relevant for estimating the total economic cost of ash dieback. Hill et al (2019, Current Biology) recently estimated the full economic cost of ash dieback in Britain at £15bn. The author of the present MS might wish to comment on how the predictions of his model would affect this estimate.

My comments below are mainly to do with clarity of presentation.

Terms like “save ash populations” and “ash populations may survive” are ambiguous. What does salvation of a population mean? Zero reduction in size? Reduction to any viable population size, however small that may be? I suggest that this could be more precisely framed.

I have changed this in the text but have modified the title along the lines suggested.

The MS could emphasize more that the simulation is over the course of one century, in a woodland where ash has to compete with other tree species. Thus, for example, the MS title might better describe the paper if it were “How many ash trees will survive ash dieback in a British woodland during the next century?”

Done as above

The approach of the SORTIE model, its parameterization on Whytham woods, and the timeframe used need to be described more fully in the MS. I suggest the abstract should mention that the model is for Whytham woods in particular, and for a 97-year timeframe. The introduction should contain a paragraph that briefly describes what the model seeks to do, how it works and what parameters it needs. It should also briefly describe the size of Whytham woods and its tree composition, and the size of the area simulated in the model.

This information now exists, there is a new paragraph in the introduction and further detail on the model is included in the penultimate paragraph of the methods.

I think that the current second and third paragraphs of the introduction could be omitted without significant detriment to the MS, as they merely give other examples of forest pests and pathogens. More relevant would be examples (if any exist) of other forest pests and pathogens for which models of impacts have been made similar to the work of the present MS. If none exist, this could be pointed out.

These have been omitted, also suggested by referee 2.

I suggest that Needham et al 2016 is discussed prior to the final paragraph of the introduction, and the final paragraph of the introduction focuses more exclusively on the work reported in the present MS.

I have modified the last paragraph to emphasise the novelty of the present work.

Heritability of resistance is an important parameter for the MS. As far as I am aware, no one has yet estimated heritability of mortality due to ash dieback. All the estimates of heritability are of ash dieback damage scores such as crown dieback and lesion length, as far as I am aware. If this is the case it would be worth noting. It is also worth noting that (as far as I am aware) all estimates of heritability so far have been in plantations with fairly uniform environments. In natural woodlands environmental variation is likely to be much greater, leading to lower heritability of resistance traits.

The referee is correct that no estimates of mortality heritability have been done – indeed it might be difficult to do so given that a tree either survives or dies, and if it dies it obviously does not pass its genes on to the next generation. The most relevant observation would be the heritability of resistance mechanisms, which have also not been studied directly. This is why as the referee correctly points out we must use heritability of damage scores.

Abstract line 20/21 “These observations have LED TO SUGGESTIONS that...”

Agreed

Page 7 line 57. With a strict regard to its Latin origin, the term “decimated” means “reduced by one in ten”.

Changed to very severely reduced

Page 15 “One feature not considered in the model presented here is that the selection differential is likely to change throughout the period considered. This is partly because intense selection will reduce the genetic variance in the offspring population (51), and also because as susceptible individuals die the proportion of the pollen in the air that originated from resistant fathers will rise and so the probability of resistant offspring being produced will increase. It is likely that both these effects will result in there being a higher population of resistant ash than suggested here.” This is an important caveat, especially if we are interested in timeframes of over a century, which deserves more emphasis.

This paragraph has been expanded somewhat, but it is not clear how emphasise it much further than has been done here.

Page 16 “The current hypothesis is that, probably in part, resistance is conferred by earlier spring leaf flushing and earlier autumn leaf senescence (15, 26)” this is only one of several hypotheses. If true, this phenology trend may simply allow escape of trees that flush and senesce earlier than other trees – if the whole population shifts to earlier flushing and senescence, this advantage may be lost.

An additional sentence has been added

Figure 1. A fuller first part of the legend would be helpful to the reader.

This has been expanded

The MS could be clearer on how the reader can access the data and code used.

There are no data in this paper, the code has been uploaded to dryad. The data used for parameterisation have already been published (Evans et al 2015, ref 39).

In future it would be useful if the author modelled the effects of management interventions like the felling of diseased trees, or the planting of “replacement” species for ash.

Journal Name: Royal Society Open Science

Journal Code: RSOS

Online ISSN: 2054-5703

Journal Admin Email: openscience@royalsociety.org

Journal Editor: Andrew Dunn

Journal Editor Email: openscience@royalsociety.org

MS Reference Number: RSOS-190908

Article Status: SUBMITTED

MS Dryad ID: RSOS-190908

MS Title: Can natural resistance save ash populations from ash dieback disease?

MS Authors: Evans, Matthew

Contact Author: Matthew Evans

Contact Author Email: mrevans@hku.hk

Contact Author Address 1: Kadoorie Biological Sciences Building

Contact Author Address 2: Pok Fu Lam Road

Contact Author Address 3:

Contact Author City: Hong Kong

Contact Author State:

Contact Author Country: Hong Kong

Contact Author ZIP/Postal Code:

Keywords: *Fraxinus excelsior*, ash, ash dieback disease, individual-based model, SORTIE, forest-gap model

Abstract: Novel pests and diseases are becoming increasingly frequent, and often cause additional mortality to host species in the newly contacted communities. This can alter the structure of the community and/or the continued presence of impacted host species. In the last twenty years ash dieback disease (ADB) has spread into Europe from East Asia. It has caused substantial mortality in ash tree (*Fraxinus excelsior*, L.) populations. However, a proportion of the individuals in most populations appears to be less susceptible to ADB and resistance seems to have high heritability. These observations have suggested that ash populations may survive the disease. In order to test this hypothesis, I modified an existing model of UK woodland to take into account the impact of ADB, and allowed offspring to inherit resistance traits from their parent. The results suggest that ash populations could be sustainable but at lower levels than they are currently. For example, when the proportion of resistant individuals is about 10% and heritability of resistance is 0.5 then the population of ash falls to about one third of present levels. The influence of the proportion of individuals initially resistant to ADB was larger than the heritability of resistance. The fact that the size of the resistant population is important to achieving a high population size in the presence of ADB suggests that a selective breeding programme with the intention of augmenting the natural ash populations would be beneficial.

EndDryadContent

Appendix C

Dear Sir,

Thank you for this good news concerning my manuscript. I have made the changes recommended by the referees as shown below. I believe that I have already provided all the other information required and the temporary dryad link provided previously is still live.

Yours truly, Matthew

Response to referees comments.

Associate Editor Comments to Author:

Thank you for making efforts to improve the paper. Only a few minor concerns remain from the reviewers - please ensure you tackle these in your revision.

Reviewer comments to Author:

Reviewer: 3

Comments to the Author(s)

The author has dealt satisfactorily with the reviewers comments. My only further suggestion is that the word "initial" is inserted before the word "proportion" on line 31 of the abstract.

Done

Reviewer: 1

Comments to the Author(s)

Thank you for your efforts in the revision.

Reviewer: 2

Comments to the Author(s)

The manuscript has significantly improved. Most of my comments and suggestions have been taken into account by the author, although sometimes somewhat desultory, but I guess that is ok. I still think that the presentation of Figure 1 needs improvement, i.e. increased font size, and figure legends would make it much easier for the reader to understand and to extract the relevant information.

I have increased the font size of the axis labels and scale

I am still confused by the calculation described in page 11. The calculated mid-points are, as I understand, the mid-points of the standard deviation range of percent damage scores of offspring of susceptible and resistant trees. If this is correct, I would expect the mid-point of offspring of resistant trees to be smaller than those of offspring of intermediate and susceptible trees. Apparently, there is something that I did not understand, but I admit that this may be my personal problem.

I think that the referee is used to seeing damage scores but the calculation is framed in terms of resistance. So high scores = high resistance which would equate to a low damage score. In other words in terms of damage score the referee is correct, but we are looking at resistance.

I could not find a statement in the discussion that a strong reduction of the host population and the number of susceptible trees is likely to cause a decrease in infection pressure.

I have added a new sentence to explain why I do not feel that it is simple to intuit an answer to this question.